# Natural History of Marburg Virus Infection to Support Medical Countermeasure Development

**DOI:** 10.3390/v14102291

**Published:** 2022-10-18

**Authors:** Jason E. Comer, Trevor Brasel, Shane Massey, David W. Beasley, Chris M. Cirimotich, Daniel C. Sanford, Ying-Liang Chou, Nancy A. Niemuth, Joseph Novak, Carol L. Sabourin, Michael Merchlinsky, James P. Long, Eric J. Stavale, Daniel N. Wolfe

**Affiliations:** 1Department of Microbiology and Immunology, University of Texas Medical Branch at Galveston, Galveston, TX 77555, USA; 2Institutional Office of Regulated Nonclinical Studies, University of Texas Medical Branch at Galveston, Galveston, TX 77555, USA; 3Battelle Biomedical Research Center, West Jefferson, OH 43162, USA; 4AmplifyBio, West Jefferson, OH 43162, USA; 5Tunnell Government Services, Inc., Supporting Biomedical Advanced Research & Development Authority (BARDA), Administration for Strategic Preparedness and Response (ASPR), U.S. Department of Health and Human Services (DHHS), Washington, DC 20201, USA; 6CBRN Vaccines, Biomedical Advanced Research & Development Authority (BARDA), Administration for Strategic Preparedness and Response (ASPR), U.S. Department of Health and Human Services (DHHS), Washington, DC 20201, USA

**Keywords:** Marburg virus, MARV, filovirus, natural history study, virus, macaques, FDA

## Abstract

The Biomedical Advanced Research and Development Authority, part of the Administration for Strategic Preparedness and Response within the U.S. Department of Health and Human Services, recognizes that the evaluation of medical countermeasures under the Animal Rule requires well-characterized and reproducible animal models that are likely to be predictive of clinical benefit. Marburg virus (MARV), one of two members of the genus *Marburgvirus*, is characterized by a hemorrhagic fever and a high case fatality rate for which there are no licensed vaccines or therapeutics available. This natural history study consisted of twelve cynomolgus macaques challenged with 1000 PFU of MARV Angola and observed for body weight, temperature, viremia, hematology, clinical chemistry, and coagulation at multiple time points. All animals succumbed to disease within 8 days and exhibited signs consistent with those observed in human cases, including viremia, fever, systemic inflammation, coagulopathy, and lymphocytolysis, among others. Additionally, this study determined the time from exposure to onset of disease manifestations and the time course, frequency, and magnitude of the manifestations. This study will be instrumental in the design and development of medical countermeasures to Marburg virus disease.

## 1. Introduction

As part of the Project Bioshield Act of 2003, the Department of Homeland Security (DHS) identified hemorrhagic fever viruses as a class A Material Threat to the population of the United States (US). Among the viruses in this class are the filovirus family, a class of filamentous RNA viruses including Ebola virus (EBOV), Sudan virus (SUDV), and Marburg virus (MARV) responsible for sporadic limited outbreaks and a high mortality rate [1]. While major advances have been made in medical countermeasures against EBOV [1,2] in the context of the recent outbreaks, medical countermeasures against MARV are much further behind. In the absence of a predictable or large outbreak, we expect any vaccine candidates against MARV will require non-traditional regulatory pathways for approval, such as the FDA Animal Rule, requiring well-characterized and reproducible animal models that are likely to be predictive of clinical benefit.

MARV was first identified in 1967 with cases in Germany and the former Yugoslavia when infected monkeys were imported from Uganda. Since that time, 12 different outbreaks have occurred with the number of cases per outbreak ranging from one to 252 [3]. Outbreaks in humans are generally initiated after spillover events from animal hosts such as the fruit bat, *Rousettus aegypticus*, as summarized with a characterization of MARV infections in a fruit bat model [4]. MARV infection causes a mild disease in the *R. aegypticus* model but causes severe disease in nonhuman primates (NHP) experimentally infected with MARV, and humans upon spillover events [4].

A 2015 review provides the most comprehensive comparison between human MARV disease and results observed in different NHP challenge models [5]. The human data are largely derived from 35 patients diagnosed with MARV infection and treated at modern medical facilities. Clinical signs generally appeared within 3 to 9 days post-exposure and progressively worsened to include diarrhea, nausea, vomiting, fever, and exanthema or enanthema. The condition of patients usually worsened during the second week, with hemorrhaging in some cases. Mortality ranged from 20−30 percent in individuals receiving intensive care, to up to 80–90 percent in rural African settings. Work from various groups has shown that the cynomolgus macaque model of MARV disease recapitulates what is seen in humans. In addition, 100% mortality is seen in animals that did not receive some type of treatment in animals challenged intramuscularly with 1000 plaque forming units (PFU). Death occurs in animals within 7−10 days (Appendix A) and is preceded by coagulopathy, fever, anorexia, changes in hematology and serum chemistry, and petechiation [6,7,8,9]. Cynomolgus macaques have been the most common animal model for evaluation of vaccines against filovirus diseases. Importantly, efficacy against disease caused by EBOV in this model correlated closely with clinical efficacy observed, both in terms of levels of protection and onset of signs of disease [10].

Here, we further characterized the cynomolgus macaque MARV intramuscular (IM) infection model by conducting a well-documented/controlled natural history study. The study was designed and executed to fulfill the FDA guidance for development of products under the Animal Rule which specifies that such studies are to be completed in advance of utilizing models for well-controlled pharmacokinetic/pharmacodynamics and efficacy studies that may be used to support product licensure.

## 2. Materials and Methods

### 2.1. Ethics Statement

This study complied with Final Rules of the Animal Welfare Act regulations (9 CFR Parts 1, 2, and 3) and Guide for the Care and Use of Laboratory Animals: Eighth Edition (Institute of Laboratory Animal Resources, National Academies Press, 2011; the Guide). This study was conducted in UTMB’s AAALAC (Association for the Assessment and Accreditation of Laboratory Animal Care)-accredited facility and was approved by UTMB’s Institutional Animal Care and Use Committee (protocol number 2006068, approved on 1 June 2020).

### 2.2. Animals

Nine male and nine female, experimentally naïve, cynomolgus macaques (*Macaca fascicularis*, Asiatic origin bred in Vietnam) weighing between 2.4−3.2 kg and 2.5−3.2 years of age were procured from Envigo (Alice, TX, USA). Prior to shipment the NHPs were verified negative for evidence of pre-existing immunity to Reston virus (RESTV), EBOV, SUDV, and MARV in addition to standard screening tests including tuberculosis, simian immunodeficiency virus, simian retrovirus 1 and 2, simian T-lymphotropic virus-1, macacine herpesvirus 1 (Herpes B virus), and *Trypanosoma*. All animals were housed in open stainless-steel standard NHP caging, provided with Certified Primate Diet (PMI, Inc., New York, NY, USA), and water was provided *ad libitum* through an automatic watering system. To promote and enhance the psychological well-being, both food and environmental enrichment were provided to each NHP.

Prior to placement on study, NHPs were surgically implanted with DST micro-T implantable temperature loggers (Star-Oddi, Gardabaer, Iceland). The temperature loggers were programmed to take measurements every 15 min. NHPs were anesthetized with Ketamine (5−20 mg/kg IM) prior to all procedures.

### 2.3. Randomization

Animals were randomized by a biostatistician to two groups: MARV-exposed (N = 12) and mock-exposed (N = 6). Groups were stratified by sex and balanced by body weight.

### 2.4. Challenge

Passage 2 of Marburg virus/*H. sapiens*-tc/AGO/2005/Angola-200501379 was obtained from BEI Resources (Lot number 200501379; Manassas, VA, USA) and was not passaged further. This well-characterized virus was diluted in Hank’s Balanced Salt Solution containing 2% heat-inactivated fetal bovine serum (HBSS/2% HI-FBS) such that animals received an intramuscular dose corresponding to 1000 PFU of MARV Angola in 0.5 mL. HBSS/2% HI-FBS was used for the mock-challenged controls. The administered challenge dose was confirmed by plaque assay of the challenge suspension collected before the challenge of the first animal and after the last animal was challenged.

### 2.5. Clinical Observation and Scoring

Animals were observed, at minimum, twice daily (6–8 h apart during the light cycle) and scored using the system shown in Appendix A. A score of 0–3 indicated that no intervention was needed; a score of ≥4 (or ≥3 in any single parameter) required additional monitoring of at least once in the evening, 4–6 h after the final late afternoon check; a score of 10 or greater is required euthanasia.

### 2.6. Euthanasia

Any animals exhibiting signs consistent with significant distress/moribundity (score of 10 or greater) were evaluated for euthanasia by a veterinarian as per the approved IACUC protocol. Animals that survived until the scheduled study termination (28 days post-challenge) were humanely euthanized. Animals that required euthanasia were sedated as previously described and euthanized by administration of a pentobarbital-based euthanasia solution (e.g., Euthasol) via intravenous or intracardiac administration according to the AVMA Guidelines for the Euthanasia of Animals.

### 2.7. Blood Collection and Processing

Blood was collected from animals on Days −4, 0, 3, 5, 7, 10, 14, 21, and 28 relative to MARV challenge and at the time of euthanasia (terminal sample). The femoral vein was used for all scheduled biosampling events and intracardiac blood was collected at terminal time points or at the end of the study (Day 28). Blood was collected into serum separator tubes and tubes containing anticoagulant, specifically ethylenediaminetetraacetic acid (EDTA) and sodium citrate. Serum was aliquoted and stored at ≤−65 °C, while blood collected for hematology and coagulation was analyzed within 8 or 2 h, respectively.

### 2.8. Anti-MARV GP IgG ELISA

Serum collected for anti-MARV GP IgG ELISA analysis was inactivated by exposure with 5 MRads of gamma radiation using a previously validated method of inactivation prior to shipment to Battelle. The ELISA was conducted essentially as described in Rudge et al., 2019 [11] using purified recombinant MARV GP with amino acid sequence corresponding to the GP from MARV Ci67 (Advanced Bioscience Laboratories, Rockville, MD, USA). Test samples were evaluated in the ELISA at a 1:50 starting dilution and reportable anti-MARV GP IgG concentrations were calculated using a qualified human serum reference standard with MARV GP-binding antibodies.

### 2.9. Clinical Chemistry

Clinical chemistry analysis was conducted on harvested serum using the Abaxis Piccolo Xpress Chemistry Analyzer in conjunction with the BioChemistry Panel Plus reagent discs to determine the levels of alanine aminotransferase (ALT), albumin (ALB), alkaline phosphatase (ALP), amylase (AMY), aspartate aminotransferase (AST), c-reactive protein (CRP), calcium (CA), creatinine (CRE), gamma glutamyltransferase (GGT), glucose (GLU), total protein (TP), blood urea nitrogen (BUN), and uric acid (UA).

### 2.10. Coagulation

Blood collected in sodium citrate tubes was used to measure Activated Partial Thromboplastin Time (aPTT) and Prothrombin Time (PT) using the IDEXX Coag DxTM Analyzer. Samples were analyzed within 2 h of collection using IDEXX Coag Dx PT and aPTT cartridges.

### 2.11. Hematology

Hematology analysis was conducted on EDTA blood using the Abaxis VetScan HM5^®^ Hematology Analyzer to measure the following parameters in whole blood collected in EDTA tubes: White blood cell concentration (WBC), lymphocyte concentration and percentage (LYM), monocyte concentration and percentage (MON), neutrophil concentration and percentage (NEU), basophil concentration and percentage (BAS), eosinophil concentration and percentage (EOS), red blood cell concentration (RBC), hemoglobin (HGB), hematocrit (HCT), mean corpuscular volume (MCV), mean corpuscular hemoglobin (MCH), mean corpuscular hemoglobin concentration (MCHC), red cell distribution width (RDW), platelet concentration (PLT), mean platelet volume (MPV), platelet hematocrit (PCT), and platelet distribution width (PDW).

### 2.12. Viral Load: Plaque Assay

Harvested serum was stored frozen (≤−65 °C) from the time of processing until plaque assay analysis. On the day(s) of plaque assay analysis, samples were thawed under ambient conditions, and assayed as described by Shurtleff et al. [12]. Crystal violet stain was used to visualize plaques and results were reported as PFU/mL of serum.

### 2.13. Viral Load: qRT-PCR

On the day of collection, harvested serum (0.05 mL) was added to TRIzol^®^ LS (5X volume; i.e., 0.25 mL) and stored at ≤−65 °C until RNA exaction. RNA was extracted from samples using the Zymo Research Direct-zol^™^ RNA MiniPrep kit (Zymo Research, Irvine, CA, USA). For sample quantification, each assay plate contained a standard curve prepared using MARV VP40 gene synthetic RNA (1.0 × 10^3^ to 1.0 × 10^10^ genome equivalents/µL [GEq/µL] in duplicate wells). For the qRT-PCR, QuantiFast Probe RT-PCR Master Mix and QuantiFast RT Mix (Qiagen, Ilden, Germany) were used in conjunction with Forward primer: 5′- CCAgTTCCAgCAATTACAATACATACA-3′, Reverse primer: 5′- gCACCgTggTCAgCATAAggA-3′ and Probe: 5′-6FAM- CAATACCTTAACCCCC-MGBNFQ-3′. Primers and probe targeted the VP40 gene from MARV (GenBank accession no. DQ447660). The qRT-PCR was conducted on a Bio-Rad CFX96^TM^ Real-Time PCR. 

### 2.14. Pathology

#### 2.14.1. Necropsy and Gross Pathology

A gross necropsy was conducted on all animals that succumbed to disease or that lived until the end of study (Day 28). Gross necropsies included examinations of the external surface of the body, all external orifices, the thoracic and abdominal cavities, and their contents.

#### 2.14.2. Tissue Collection for Histopathology

The following tissues from all animals were collected during necropsy and placed in 10% neutral buffered formalin for at least 21 days: lymph nodes (axillary from infected arm, mediastinal, mesenteric, and inguinal), challenge site (skin and underlying muscle), adrenal glands, stomach with pyloris, jejunum, duodenum, ileum, transverse colon, rectum, gall bladder, liver, spleen, kidneys, heart, lungs, and any gross lesions. All tissues were processed to slides and stained with hematoxylin and eosin and evaluated by a Board-Certified Veterinary Pathologist. Histopathologic grades were assigned according to the following scale: minimal, mild, moderate, and marked (Appendix A).

### 2.15. Statistical Analyses

All statistical analyses were conducted using SAS^®^ (version 9.4) on the 64-bit platform or R. (version 3.6.3). All results are reported at the 0.05 level of significance. Mortality rates and exact 95% confidence intervals were calculated for each group using Clopper-Pearson 95% confidence intervals. A one-sided Boschloo’s test was performed to assess whether mortality outcome in the MARV-exposed group was greater than that of the mock-exposed control group. Time to death data were analyzed and compared between the MARV-exposed and the mock-exposed control groups. Kaplan–Meier curves were plotted, and median times to death were estimated with 95% confidence intervals. A log rank test was used to determine if there was a significant difference between the groups. All animals were included in the analysis; those surviving to end of study (termination on Day 28) were censored at the time of terminal sacrifice.

The maximum daily clinical scores for each study animal were used in the analysis. Summary statistics including means, standard deviations, minimums, medians, and maximums were calculated by group and study day. Median clinical scores were plotted by group and study day. In addition, the final clinical scores were plotted. For animals found dead, the final clinical score was the most recent clinical score, which may have been collected the previous day.

Analysis of variance (ANOVA) models were fitted separately to each clinical chemistry, hematology, and coagulation parameter with effects for group, study time point, and the interaction between group and study time point used to assess the model assumption of normality and to identify potential outliers. Deleted studentized residuals were calculated for each observation. If the absolute value of the deleted studentized residual was greater than 4, then the observation was considered a potential outlier. Each animal’s Day 0 value served as its own baseline in the shift from baseline analysis.

## 3. Results

### 3.1. Study Design

The objective of this study was to characterize the Marburg virus (MARV) disease in cynomolgus macaques. Eighteen (18) cynomolgus macaques were randomized to a MARV-exposed group (n = 12) and a mock-exposed control group (n = 6). The challenged animals were exposed to a targeted 1000 PFU dose of MARV Angola via intramuscular injection while the controls received vehicle only. The challenge dose was titrated on day of challenge to 6.50 × 10^3^ and 8.00 × 10^3^ PFU/dose for the pre- and post-challenge aliquots, respectively.

### 3.2. Mortality and Humane Scoring

All MARV-exposed animals were euthanized or succumbed to disease; there were no survivors after Day 8 (Figure 1). The mock-exposed NHPs lived to the end of the study (Day 28). A one-sided Boschloo’s Test showed a significant difference in mortality between the two groups (*p* < 0.0001). The median time to death for the MARV-exposed NHPs was 175.55 h (7.3 days).

For MARV-exposed animals, the first clinical scores above 0 were observed on Day 5 and clinical scores increased through Day 8, by which time all MARV-exposed animals had succumbed to disease (Appendix A). All challenged NHPs showed clinical signs characteristic of MARV disease including anorexia, hunched posture, and hemorrhage (petechiation) which continued to worsen until the animal met euthanasia criteria or succumbed to infection. Clinical scores remained at 0 for all mock-exposed control animals throughout the study (Figure 2) (Appendix A).

### 3.3. Viremia

#### 3.3.1. qRT-PCR

On Day 3 post-challenge, 2/12 MARV-exposed NHPs were positive for MARV RNA in serum as measured by qRT-PCR (mean 2.82 GEq/µL). On Day 5, all (12/12 NHPs) were positive for viral RNA in serum. On Day 5, the mean titer was 5.72 × 10^6^ GEq/µL and increased to 8.59 × 10^6^ GEq/µL on Day 7 (Figure 3, Appendix A). The mean titer was 8.13 × 10^8^ GEq/µL at the terminal.

#### 3.3.2. Plaque Assay

Levels of viable MARV in the serum were determined by plaque assay. Viable virus titers are illustrated in Figure 4. Interestingly, 12/12 MARV-exposed NHPs showed detectable levels of viable virus in circulation on Day 3 compared with 2/12 positive for serum MARV RNA by qRT-PCR. The mean titer on Day 3 was 1.48 × 10^3^ PFU/mL, which increased to 6.35 × 10^7^, 8.06 × 10^7^ and 8.81 × 10^7^ PFU/mL on Days 5, 7, and at the terminal timepoint, respectively (Appendix A).

### 3.4. Body Temperature

Body temperatures were recorded every 15 min via implantable loggers which were removed during necropsy and data downloaded (Figure 5). There were many statistically significant increases and decreases from baseline body temperature for both MARV-exposed animals and mock-exposed control animals as depicted in Figure 5. The temperature shifts were mostly below the baseline on Days 1 and 2 and mostly above the baseline between Days 3 and 8 for MARV-exposed animals. The MARV-exposed NHPs showed a marked increase in body temperature ranging from 1–3.5 °C above baseline starting on Day 3 post-infection. The increase in body temperature continued until the animal met euthanasia criteria or succumbed to infection.

### 3.5. Clinical Chemistry and Hematology

There were several changes in clinical chemistry and hematology parameters that are characteristic in both MARV infected humans and NHPs. Statistically significant changes were noted for numerous clinical chemistry and hematology parameters at days 3, 5, and 7 post exposure in MARV-exposed NHP compared to mock-exposed controls (Figure 6) and relative to baseline (Appendix A, Appendix A). As shown in Figure 6, GLU, CA, ALB, TP, and AMY levels in MARV-exposed NYP were significantly lower than those in mock-exposed controls at all three days, while ALT, AST, ALP, CRP, and GGT were significantly greater than control at days 5 and 7. Fewer changes in hematology parameters were noted, but LYM levels were significantly lower and NEU levels significantly greater in MARV-exposed NHP than mock-exposed controls at days 3 and 5 post-exposure.

There were significant increases in ALT levels from baseline (Day 0) in the mock-exposed control group. The geometric mean on Day 0 was 27.93 U/L and increased to 34.17 and 34.11 U/L on Days 3 and 10, respectively which were still in the normal ranges. There were also significant increases as a proportion of baseline for the MARV-exposed group. The geometric mean level of ALT in the MARV-exposed group was 38.25 on Day 0 which increased to 43.32, 254.81, 820.48, and 1187.59 U/L on Days 3, 5, 7 and at the terminal timepoint (Appendix AA). The MARV-challenged NHPs had significantly higher levels of ALT than mock-exposed controls at Days 5 (*p* = 0.0005) and 7 (*p* < 0.0001) (Figure 6).

There were also significant increases in AST levels above baseline for the MARV-exposed group (Appendix AB). On Day 0, challenged NHPs had a geometric mean level of 39.21 U/L which increased to 624.22, 1811.12, and 1961.98 U/L on Days 5, 7, and at the terminal timepoint, respectively. The levels of AST were significantly higher in the MARV-exposed NHPs compared to the mock-exposed controls on Days 5 and 7 (both *p* < 0.0001) (Figure 6).

ALP and GGT were also dramatically increased in the MARV-exposed NHPs at Days 5, 7 and the terminal timepoint (Appendix AH,I). Decreases in GLU and ALB were also observed providing further evidence for liver damage.

The GLU levels are illustrated in Appendix AC. There was a significant increase from baseline (64.25 mg/dL) for the MARV-exposed group on Day 3 (68.58 mg/dL). However, the GLU levels began to decrease as the disease progressed and significant decreases from baseline were observed on Days 7 (26.38 mg/dL) and at the time of death (21.56 mg/dL). There were significant group effects on Days 3 (*p* = 0.0154), 5 (*p* < 0.0001), and 7 (*p* < 0.0001) (Figure 6).

Changes were also seen in ALB levels over the course of study (Appendix AD). MARV-exposed NHPs showed a consistent decrease in ALB level until death. There were significant decreases from the baseline mean of 3.36 mg/dL to 3.07, 2.75, and 2.21 mg/dL on Days 5 and 7 and at the terminal timepoints, respectively. The levels were significantly different between the two groups on Days 3 (*p* < 0.0001), 5 (*p* < 0.0001), and 7 (*p* < 0.0001) (Figure 6).

Along with liver damage there is also evidence of kidney involvement, with increased levels of both BUN (Appendix AE) and CRE (Appendix AF) for MARV-exposed NHPs at the time point prior to and at the time of euthanasia/death.

There was a significant decrease in BUN compared to baseline in the MARV-exposed group on Day 5 but a significant increase on Day 7 and at the terminal timepoint. The challenged NHPs had a mean baseline level at 20.01 mg/dL on Day 0 which decreased to 17.21 on Day 5 then increased to 28.61 on Day 7 and 49.42 at the terminal timepoint. Significant differences in BUN levels between the two groups was observed, with lower BUN levels in the MARV-exposed group on Days 3 (*p* = 0.0015) and 5 (*p* = 0.0004), and higher levels on Day 7 (*p* = 0.0267) (Figure 6).

There were also significant increases as a proportion of baseline in the CRE levels for the MARV-exposed group on Day 7 and at the terminal timepoint (Appendix AF) but there were no group differences observed (Figure 6).

There were significant increases in CRP levels from baseline for the MARV-exposed group on Days 3, 5, 7, and at the terminal timepoint (Appendix AG). The CRP levels on Day 0 were 3.71 mg/L but increased to 7.82, 35.81, 24.54, and 21.45 mg/L on Days 3, 5, 7, and the terminal timepoint, respectively. There were significant group effects on Days 3 (*p* = 0.0087), 5 (*p* < 0.0001), and 7 (*p* < 0.0001) (Figure 6).

There was no significant difference in WBCs between the MARV and mock-exposed groups (Figure 6). There was a significant decrease as a proportion of baseline for the mock-exposed group on Day 14 (Appendix AA). There was a significant increase as a proportion of baseline for the MARV-exposed group on Day 7.

MARV-exposed NHPs presented with transient lymphopenia following challenge which resolved just prior to death (Appendix AB). There were significant decreases from baseline on Days 3 and 5. There was a significant increase as a proportion of baseline for the mock-exposed group on Day 21. There were significant group effects on Days 3 (*p* = 0.0016) and 5 (*p* = 0.0006) (Figure 6).

MARV-exposed NHPs also presented with a transient neutrophilia that, like the decrease in lymphocytes, resolved at the terminal time point (Appendix AC). There were significant increases from baseline for the MARV-exposed group on Days 5 and 7. There was a significant decrease from baseline for the mock-exposed group on Days 14 and 21.

MARV-exposed NHPs showed a decrease in PLTs after challenge and there were significant decreases from baseline on Days 5 and 7 and the terminal timepoint (Appendix AD). PLT counts were significantly lower in the challenged animals on Days 3 (*p* = 0.0153) and 5 (*p* = 0.0215) compared to mock-exposed controls (Figure 6).

### 3.6. Coagulation

There were significant increases as a proportion of baseline of aPTT time for the MARV-exposed group on Study Day 7 and at Terminal (Appendix A). The aPTT time on Day 0 was 77.81 s, this increased to 84.48 s on Day 5, 204.78 s on Day 7, and 265.28 s at the time of death/euthanasia (Figure 7A). There was a significant group effect on Day 7 (*p* < 0.0001) (Figure 7A). After removing potential outliers there was a significant increase as a proportion of baseline for the MARV-exposed group on Day 5.

There were significant increases from baseline for the MARV-exposed group on Days 5, 7 and at the terminal timepoint where the PT time extended from 21.27 to 33.42, 62.67, and 94.49, respectively (Figure 7B). There were significant group effects on Days 5 (*p* = 0.0016) and 7 (*p* < 0.0001) (Figure 7B).

It is important to note that at the terminals time point, only two NHPs in the MARV-exposed group clotted within the 300 s time limit for the aPPT assay and 100 s limit for the PT. The out-of-range samples were assigned 300 (aPTT) or 100 (PT) seconds for statistical analysis and graphing purposes.

### 3.7. Pathology

#### 3.7.1. Macroscopic Findings

In the MARV-exposed animals, gross lesions consisted of discoloration of hearts, kidneys, and adrenal glands. Livers were diffusely pale yellow to grey. Lungs and lymph nodes were diffusely dark red to black. Lymph nodes were also enlarged. The jejunum of one animal was thickened and dark colored. Skin petechiation was a common finding. Petechial rash occurred most frequently on the thorax and at the injection site on the forearm, but was also noted on the face, inguinal area, and perineum. The mucosa of the stomach and urinary bladder was discolored dark brown and red respectfully in separate animals. Discolorations of various organs were attributed to MARV induced vascular congestion or hemorrhage seen on microscopic examination.

#### 3.7.2. Microscopic Findings in NHPs That Succumbed to Infection

Microscopic lesions in MARV-exposed animals consisted of minimal to moderate lymphocyte apoptosis in various organs, especially prominent in lymph nodes and spleen (Figure 8). Unlike mock-exposed controls which had minimal apoptosis in germinal centers due to immune stimulation, MARV-exposed animals had minimal to moderate apoptosis which extended throughout the lymphoid tissue or organ. In spleen, apoptosis was noted throughout both red and white pulp but was most prominent in white germinal centers and mantle zones. In lymph nodes, there was apoptosis in germinal centers, the mantle zone, medullary cords, and sinuses. Apoptosis in the gastrointestinal system was most prominent in Peyer’s patches and lymphofollicular tissue but extended to lymphocytes in the lamina propria. Apoptotic lymphocytes were also noted in kidney and at injection sites.

Livers were congested and there was scattered inflammation consisting of lymphocytes, macrophages, and few neutrophils. Similar inflammatory cells and minimal cellular debris were present throughout hepatic sinuses. Rare individual and clusters of apoptotic hepatocytes were occasionally observed. There were low numbers of indistinct intracytoplasmic eosinophilic inclusions within hepatocytes throughout the hepatic parenchyma.

There was mild inflammation consisting of lymphocytes and macrophages, hemorrhage, and apoptotic debris in the subcutis and deep muscle of injection sites. Congestion seen in various organs was due to vascular stasis.

### 3.8. GP IgG ELISA

No NHP on study had detectable levels of anti-MARV GP IgG antibodies. All MARV-exposed NHPs succumbed to infection prior to prompting a measurable humoral response to the virus. Importantly, there was no cross infection of the mock- exposed controls as evidenced by the lack of seroconversion.

## 4. Discussion

The development of effective countermeasures against rare diseases with high mortality and consequence is the responsibility of the Biomedical Advanced Research and Development Authority (BARDA). The development of such medical countermeasures relies on the evaluation of efficacy using animal models following the precepts of the FDA Animal Rule. The FDA Animal Rule was created to address the difficulty in developing medical countermeasures against diseases where clinical trials to demonstrate efficacy are either not feasible or not ethical. The central underpinning of efficacy evaluation is the use of reproducible, predictive animal models to demonstrate countermeasure efficacy and establish dose. This manuscript describes a BARDA-supported effort to determine if an infection of cynomolgus macaques can provide a model for evaluation of Marburg virus countermeasures. In this study, 12 animals were infected intramuscularly with a targeted 1000 PFU of MARV, and changes in clinical parameters were compared to 6 unchallenged time-matched controls as well as their own baseline data. There were two main aims with this experimental design. One aim was to determine, through the measure of physiological parameters (serum chemistry, hematology, coagulation times, body temperature) and viremia, if the infection was reproducible and could provide a faithful model for countermeasure evaluation. The second objective was to see if the kinetics of infection caused changes from baseline values that could serve as triggers for medical intervention. The data from this study confirm the cynomolgus macaque model as an appropriate model for evaluation of MARV countermeasures and the clinical presentation is reproducible when compared to published studies [5,6,7,8,9]. The rapid time course of the disease in this model after onset of clinical signs makes the identification of triggers for medical intervention problematic, implying that time-based intervention, as used for the models to evaluate Ebola countermeasures, is most appropriate [13,14,15,16,17,18,19]. Development of medical countermeasures against filoviruses and other high-consequence pathogens poses an ethical challenge to the traditional pathway to regulatory approval through demonstration of efficacy in well-designed and monitored clinical trials. The FDA Animal Rule provides the foundation for an alternative regulatory pathway by use of an appropriate and well-characterized animal model with the following requirements: (1) a reasonably well-understood mechanism of pathogenesis and its prevention or substantial reduction by the countermeasure, (2) demonstration of the effect of the countermeasure in one or more animal species sufficiently well characterized to predict the response in humans, (3) the animal study endpoint to be clearly related to the desired benefit in humans, and (4) sufficient data to select an efficacious dose in humans. Given the possibility that prophylactics and therapeutics developed against MARV may be licensed under the Animal Rule or the accelerated approval pathway, BARDA chose to develop a robust challenge model with uniform lethality that can provide conservative estimates of effectiveness for bridging of non-clinical immunogenicity and survival data to human immunogenicity data.

The development of products effective against EBOV used animal models with a 1000 PFU intramuscular challenge to demonstrate efficacy and establish the appropriate dose to establish a severe challenge model [20,21]. Our primary goal in this study was to determine if an analogous model using MARV as the challenge agent was similarly adaptable for evaluation of countermeasures. Analogous to the observation in cynomolgus macaques infected with either Ebola or Sudan filoviruses [20,22,23], a short refractory period with few signs of disease was followed by increasing clinical scores starting at Day 5 post challenge for some animals and increasing rapidly for all infected animals on subsequent days. The median time to death was 7.3 days post-challenge.

One of the traits of filovirus infection is a sustained fever. In the current study, elevated temperature was detected between day 3 and day 4 post-challenge in all infected animals and continued until very late in infection, where a loss of thermal control resulted in a dramatic temperature drop. This loss of thermal regulation and rapid drop in temperature was observed in 8 of 10 animals at time of euthanasia. The two animals that succumbed to disease had a precipitous temperature drop overnight, although it is difficult to separate the temperature drop that occurred prior to death due to loss of thermal control with the drop in temperature that occurred after death. In this study, the temperature was obtained post in-life phase by downloading the information from temperature loggers. The monitoring of temperature in real-time may provide additional data to establish appropriate euthanasia criteria in future studies.

The description of MARV infection in human and nonhuman primates can be found in Glaze et al. [5]. In the study described in this manuscript, changes in clinical chemistry and hematologic and coagulation times were used to demonstrate that the infection of cynomolgus macaques with this preparation of MARV Angola at 1000 PFU faithfully recapitulates the disease characteristics observed in human infections. In every infected animal, evidence for severe Marburg disease became clear at later times post infection with the levels of enzymes indicative of systemic inflammation, liver damage, coagulopathy, and necrosis increasing at or later than 5 days after infection. These symptoms are observed in filovirus infection of humans, suggesting that the infection in the nonhuman primate resembles the human infection sufficiently well to provide a model which will be predictive of clinical benefit for the evaluation of medical countermeasures.

The presence of circulating MARV was determined in two ways: blood samples were screened for infectious virus by plaque assay and for the level of genomic RNA by qRT-PCR. A low level of infectious virus was detectable in all infected animals on Day 3 post-challenge with a rapid increase in viral titer until euthanasia. The presence of RNA copies of the MARV VP40 gene was measured by qRT-PCR on RNA extracted from blood draws. Interestingly, the first evidence for MARV RNA was detected only in two animals on Day 3 post-challenge and on Day 5 post-challenge for the other 10 infected animals. The Day 3 plaque and qRT-PCR results suggests the plaque assay is more sensitive in detecting viremia. It is important to note that neither assay has undergone a formal qualification or validation. Future work will investigate the sensitivity of the qRT-PCR assay for use on Animal Rule studies. However, both measures of viral load rapidly increased at each subsequent blood draw until euthanasia was reached.

The goal of this study was to demonstrate the feasibility of an NHP model for the evaluation of MARV countermeasures using cynomolgus macaques infected with 1000 PFU of MARV Angola via intramuscular injection. This study described the survival, clinical chemistry, hematology, pathology, coagulation, and viremia in infected animals while determining the time from exposure to onset of disease manifestations and the time course, frequency, and magnitude of the manifestations. Evidence of liver damage was apparent by the significant increases in ALT and AST in MARV-infected animals over the mock-exposed group by Day 3. Evidence of kidney damage was also apparent with increased levels of both BUN and CRE.

The portfolio of products to address a MARV outbreak should include both prophylactic vaccines and therapeutic antivirals to provide maximum outbreak control. The model as presently designed in this study is suitable for evaluation of vaccine efficacy with immunization and subsequent challenge. The data from this study provide many candidate signs of disease that can be monitored for evidence of viral infection. The primary endpoint will be survival but secondary endpoints measuring the amount of breakthrough can utilize many of the virus induced physiological changes described in this study. The evaluation of therapeutic countermeasures requires delayed medical intervention until an infection is well established to mimic the stage of infection representative of that encountered in the clinic. The evaluation of some therapeutic countermeasures using the Animal Rule has utilized biomarker detection as a sign for medical intervention [24], while other models have relied on time-based intervention [25]. In this model, the rapid disease progression and absence of an easily detectable biomarker unique for MARV infection prior to Day 5 after challenge, suggests a time-based intervention will be most feasible. The high dose and infectivity of MARV Angola [9] contribute to the rapid disease course and high mortality in the model, but the evaluation of countermeasures in severe challenge models ensures that the countermeasures that succeed in such models will be effective in the field when used to treat this disease with high mortality in humans.

## Figures and Tables

**Figure 1 viruses-14-02291-f001:**
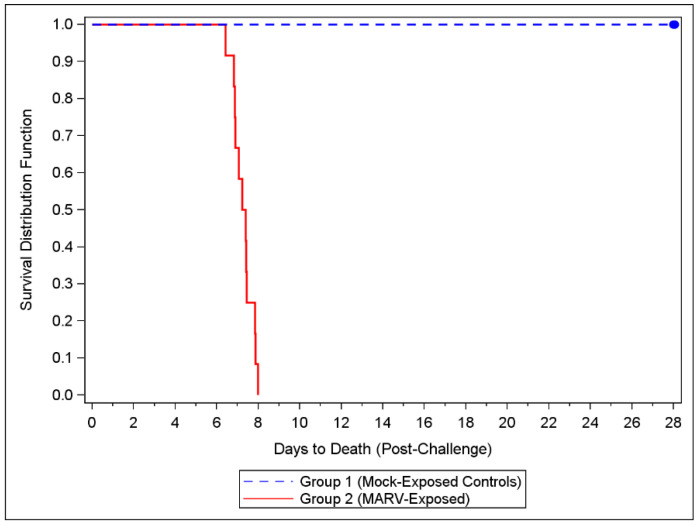
Kaplan-Meier Time to Death Plot. Twelve cynomolgus macaques were challenged with targeted dose of 1000 PFU of MARV Angola via IM injection. An additional six NHPs were injected with HBSS/2% HI-FBS and served as mock-challenge controls. All NHPs were observed at least twice daily for up to 28 days post-challenge.

**Figure 2 viruses-14-02291-f002:**
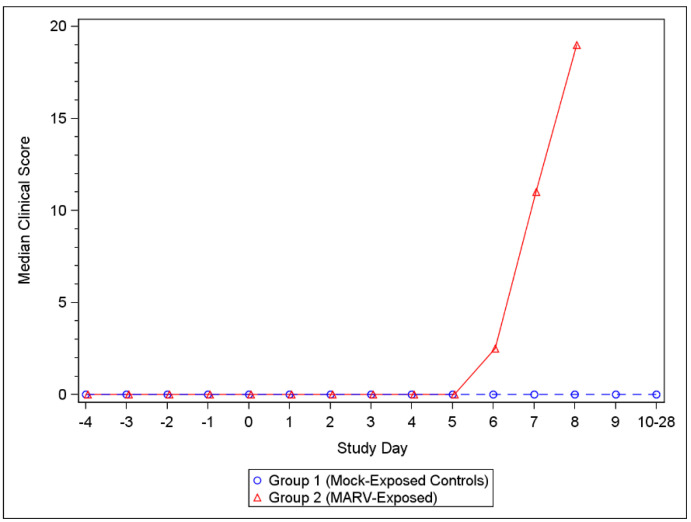
Median Clinical Scores for Mock-exposed and MARV-exposed animals. Following challenge, NHPs were scored for clinical presentation of disease at each observation. A score of 0–3 indicated that no intervention was needed; a score of ≥4 (or ≥3 in any single parameter) required additional monitoring; a score of ≥10 required euthanasia. See Appendix A for the scoring criteria, and Appendix A for individual clinical scores.

**Figure 3 viruses-14-02291-f003:**
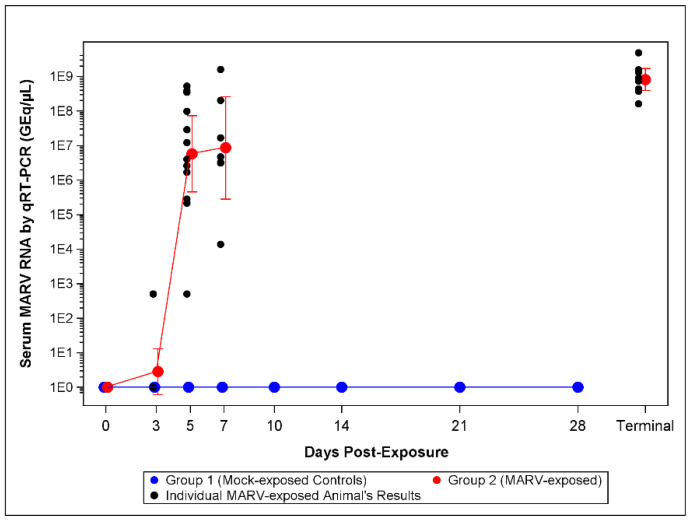
qRT-PCR Assay Serum MARV RNA Titers. The lower limit of quantitation (LLOQ) for the assay is 1.0 × 10^3^ GEq/µL and the lower limit of detection LLOD is 0. For statistical analysis and graphing LLOQ samples were assigned 5 × 10^2^ GEq/µL and LLOD samples were assigned 1.0 GEq/µL. Symbols represent geometric mean concentrations and bars indicate 95% confidence intervals. The confidence intervals on all Days 0 through 28 for mock-exposed control animals, and on Day 0 for MARV-exposed animals, were not plotted as the results were all below the limit of detection for the assay. Terminal draws in animals that met euthanasia criteria were plotted as a single time point.

**Figure 4 viruses-14-02291-f004:**
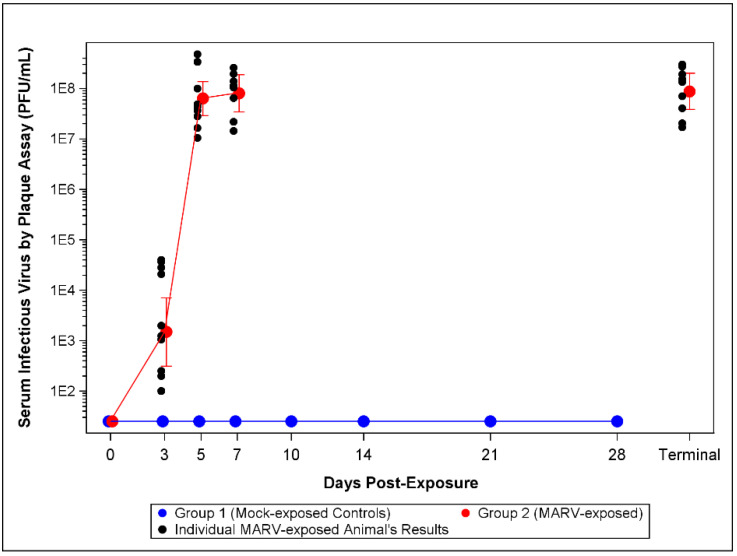
Plaque Assay Viremia Titers. The LLOD for the assay is 50 PFU/mL. For statistical analysis and graphing LLOD samples were assigned 25 PFU/mL. The symbols represent geometric mean concentrations and bars indicate 95% confidence intervals. The confidence intervals on Days 0 through 28 for mock-exposed control animals, and on Day 0 for MARV-exposed animals were not plotted as the results were constant and all below LLOD of 50 PFU/mL.

**Figure 5 viruses-14-02291-f005:**
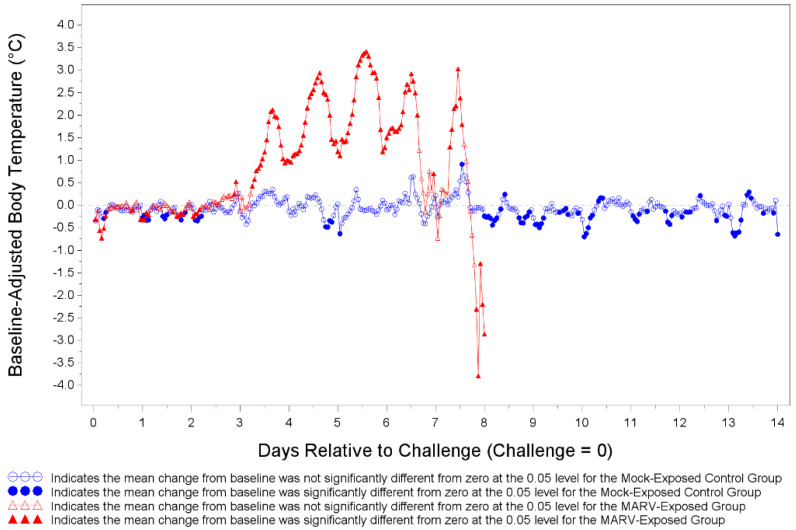
Least Square Mean Plot for Shift from Baseline Body Temperature (°C) for Mock-exposed and MARV-exposed Animals. Prior to challenge all NHPs were implanted with DST micro-T temperature loggers. Body temperature was recorded every 15 min. Body temperature was analyzed based on hourly averages of the pre-challenge baseline and baseline-adjusted post-challenge data. Baseline-adjusted post-challenge hourly averages were used to correct for circadian or diurnal rhythms. Days 15 through 28 for mock-exposed animals are not plotted.

**Figure 6 viruses-14-02291-f006:**
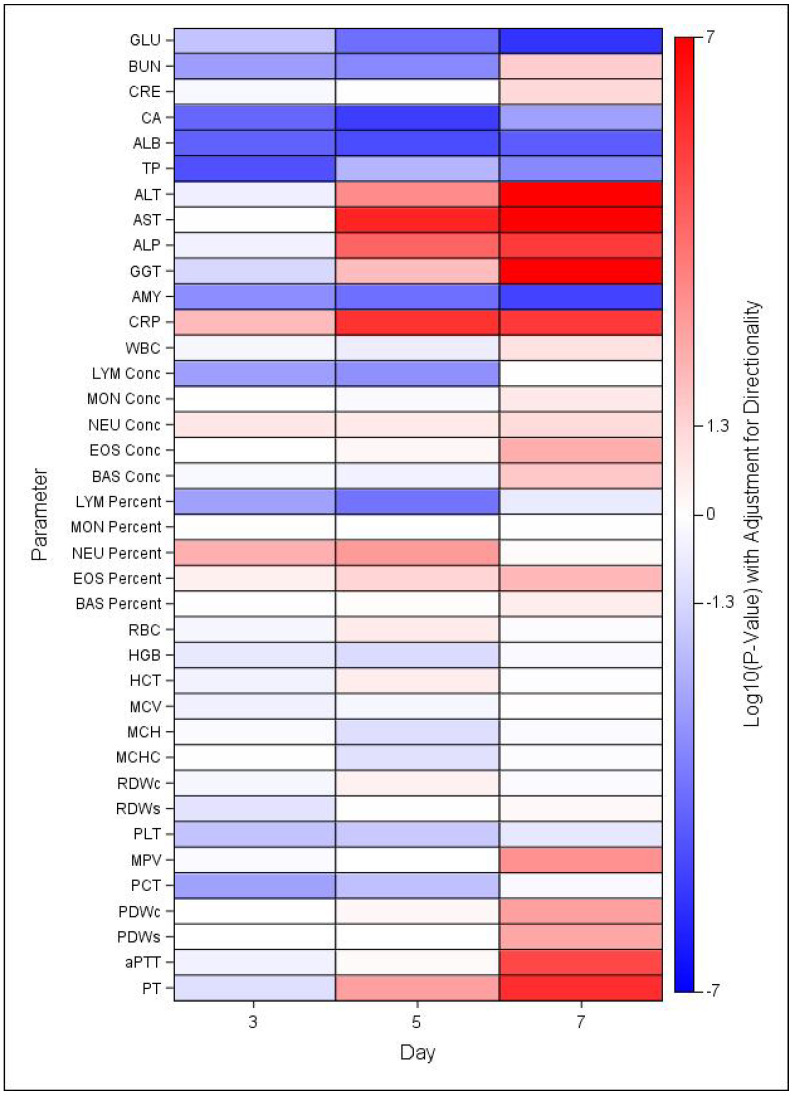
Heat Map of Statistical Differences between the MARV-Exposed and Mock-exposed Groups. Heatmap indicating mean increases (red) or decreases (blue) in MARV-exposed group in comparison to mock-exposed control group, based on log-transformed *p*-values adjusted for directionality. Significant differences indicated for values >1.3 or <−1.3, where log10(0.05) = 1.3.

**Figure 7 viruses-14-02291-f007:**
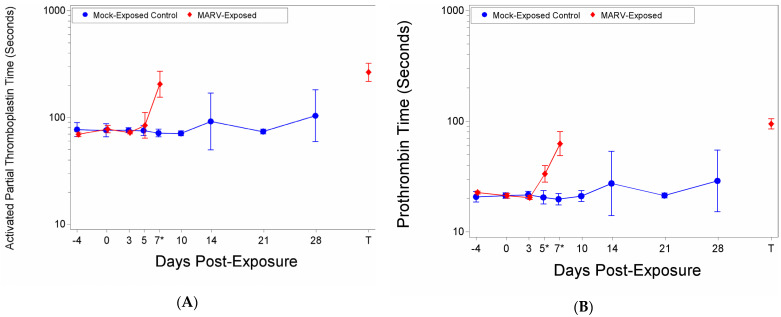
Coagulation Factors. (**A**) Activated Partial Thromboplastin time; (**B**) Prothrombin time. At each blood draw coagulation times were measured using a IDEXX Coag DxTM Analyze and IDEXX Coag Dx PT and aPTT cartridges. The symbols represent geometric mean times and bars indicate 95% confidence intervals. * indicates statistically significant difference in means (*p* < 0.05) between MARV-exposed and mock-exposed control groups.

**Figure 8 viruses-14-02291-f008:**
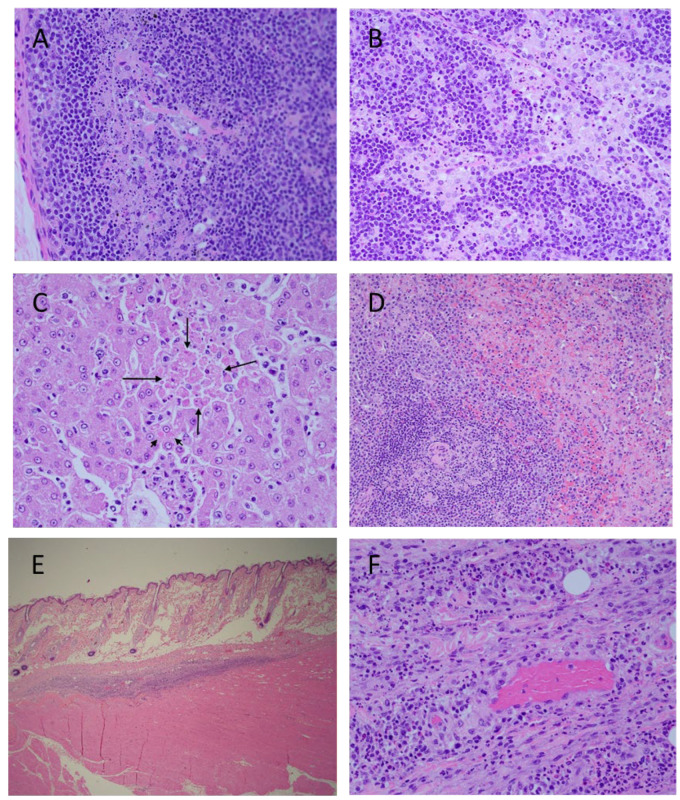
Histopathology Findings in MARV-exposed Cynomolgus Macaques. Cortex (**A**) and medulla (**B**), inguinal lymph node, extensive lymphocyte apoptosis. Liver (**C**), hepatocyte apoptosis (long arrows) and hepatocyte intracytoplasmic inclusion bodies (short arrow). Spleen (**D**), extensive lymphocyte apoptosis in white pulp and red pulp. Challenge site with mild inflammation (**E**). Higher magnification of challenge site (**F**), apoptosis of inflammatory lymphocytes. Photographs are representative.

## Data Availability

The data are available upon request.

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
