# Peer review of "Natural History of Marburg Virus Infection to Support Medical Countermeasure Development"

_viruses, 2022, doi:10.3390/v14102291_

Round 1

Reviewer 1 Report

Comments are in the PDF of the manuscript. Overall, the manuscript needs to demonstrate more comparison to existing data for MARV infection of NHPs. It doesn't explicitly describe how their data compare. 

Reviewer 2 Report

In the manuscript entitled “Natural history of marburgvirus infection to support medical countermeasure development,” the authors present a summary of the data collected from an experiment in which 12 cynomolgus macaques were inoculated with 1000 PFU of Marburg virus (MARV) variant Angola and observed until they reached the humane endpoint. The authors compare trends in various biomarkers (including viremia and blood chemistry parameters) from the MARV-exposed animals to 6 uninfected control animals as well as to baseline levels. The results of this natural history study agree with previous studies, demonstrating that, at the dose level used, MARV/Ang causes severe and uniformly lethal disease in nonhuman primates that closely resembles what is observed in lethal human disease. The goal of this study was to provide a well-characterized and reproducible animal model for MARV that would fulfill FDA requirements for countermeasure development under the Animal Rule. Although the study achieves this goal superficially, there are several missing pieces of information that should be added to create a robust and meaningful data set that can be used to inform future experiments with this model system. To this end, several concerns are outlined below.

Major Concerns

1. On Lines 255-256, the authors mention that all challenged NHPs “showed clinical signs characteristic of MARV disease.” Can the authors describe these clinical signs in detail for each animal over the course of the experiment? These details could be provided in a table, which would be a useful resource for understanding this model system and comparing it to previous experiments or future experiments performed by other investigators.

2. I would strongly urge the authors to provide data for individual animals for all of the parameters assessed throughout the course of the experiment. For instance, it would be very useful to be able to track the following data points for each animal over the course of the study: (1) sex, (2) age and weight at outset, (3) clinical score, (4) time of death, (5) virus RNA levels in the blood, (6) infectious virus levels in the blood, (7) average daytime body temperature, (8) select hematology and blood chemistry parameters, (9) coagulation times. Such data could be organized in a table or spreadsheet and added to the supplemental information. In the current version of the manuscript, only mean values are provided for each parameter analyzed, and it is therefore impossible to interrogate the relationship between parameters in any given animal. Providing these data would provide an additional layer of detail that could be invaluable to others in the field, as well as to the FDA, when/if it comes time to evaluate a product based on the Animal Rule. Moreover, since the authors state that one aim of this study was to provide biomarkers of disease (Lines 455-457 and 530-533), it would be helpful if they could provide detailed information regarding these biomarkers.

3. The Abstract indicates that the animals were observed for body weight, but this data is not included in the manuscript. Can the authors provide these data (or explain why they are not relevant)?

4. The Materials and Methods section indicates that the anti-MARV GP antibody response was assessed in the animals; however, the results are not described (presumably because no antibody response was mounted in the ~7 days prior to the death of the MARV-exposed animals). Please describe these results.

5. Did the authors quantify the levels of virus RNA or infectious virus in tissues at endpoint? If not, why not? If so, please include these data.

6. Cytokines are considered to play a role in the pathogenesis of filovirus diseases, including MARV disease, and have often been examined in filovirus studies in animal models. Considering that one of the stated aims of this paper is to identify biomarkers of disease, did the authors investigate the cytokine response? Please also include a discussion of the current understanding of the roles of cytokines in MARV infection.

7. How were the baseline values calculated for the clinical chemistry, hematology, and coagulation parameters? Were they based on the average values from all animals (i.e., MARV-exposed and mock) at 0 DPI? Line 316 implies that the baseline values may have been calculated using only the data from the MARV-exposed animals, but they ought to have been calculated using the data from the mock animals as well. Please describe this in the Materials and Methods section. If possible, please also indicate the baseline for each parameter in each figure (e.g., with a dashed line).

8. How were the normal ranges calculated for the clinical chemistry parameters? Please describe this in the Materials and Methods section. If possible, please also indicate the normal ranges for each parameter in each figure.

9. Please comment on the degree to which the histology findings (Section 3.7.2) are applicable (or not) to all MARV-exposed animals? Were samples from all animals evaluated by histopathology? Or were only a subset of animals assessed? Please describe this in the Materials and Methods or Results Sections and indicate that the images shown in Figure 8 are representative (if applicable). Were the tissues scored in any way (e.g., to provide a semi-quantitative assessment of degree of apoptosis or inflammation)? If so, please provide these scores.

10. The majority of the figures lack figure legends. The authors should provide detailed figure legends for each figure; this would relieve the reader of having to search through various parts of the manuscript text to find the details relevant to each figure (i.e., description/definition of axes, statistical tests performed, n, etc.).

11. Please move Figure 6 to the supplement and move Figures S1 and S2 into the main body of the manuscript. The authors dedicate almost the entirety of Section 3.5 to describing specific blood chemistry and hematology parameters that are depicted in Figures S1 and S2; all of these parameters are important indicators of filovirus disease and deserve to be highlighted in the main body of the text. Meanwhile, the relevance of Figures 6 is unclear: The figure itself is poorly described in the manuscript and rarely cited; many of the parameters depicted in this figure are not mentioned at all (other than in the Materials and Methods); it is impossible to determine which values are significant or not when the Log10 P-value is around 1.3 or -1.3; and the results themselves seem secondary to what is already described/depicted for Figures S1 and S2.

12. Please also include the data for ALP and GGT in Figure S1.

13. Is the median of the daily clinical scores the most appropriate value to report in Figure 2? The authors mention that the first clinical scores above “0” were observed on Day 5, but by expressing median values, this statement is not reflected in Figure 2 (for which the median clinical score at Day 5 is “0”).

14. Filovirus nomenclature should be properly used in the Introduction (Lines 44-45). For example, Ebola virus (abbreviated “EBOV”) is the type virus belonging to the species Zaire ebolavirus (which should not be abbreviated as “EBOV”). Likewise, it would be more accurate to replace “Marburgvirus” in the manuscript title with “Marburg virus”.

15. The summary of other well documented studies on the pathogenesis of MARV infection in nonhuman primate models is missing a few key papers, including Lisa E. Hensley, et.al., 2011 (PMID: 21987738) and Timothy K. Cooper, et.al., 2018 (PMID: 30053050).

Minor Concerns

 1. Please indicate the route of MARV inoculation in the Materials and Methods section. Although this information is provided in the Discussion section (Line 520), it should also be described in Section 2.4.

2. Can the authors provide passage information and sequence confirmation for the MARV isolate used to inoculate the animals? Was the virus stock received from BEI amplified before it was used in this study?

3. In Section 2.9, please provide the abbreviations for all parameters that were assessed (i.e., add GLU, CRE, ALB, etc.).

4. The authors should consider adding a brief paragraph at the outset of the Results section to setup the experimental outline. While this information can be gathered from the Materials and Methods section, the added context in the Results section would be helpful.

5. It is an interesting observation that only two of the MARV-exposed animals showed detectable viral RNA in the blood despite the fact that all of the animals had detectable infectious virus in the blood at the same time point. The authors mention these results again in the Discussion section but they offer no possible reasons for the discrepancy. Please revise the Discussion to include some reasons for this observation.

6. The quantification of MARV RNA levels in the blood is not described consistently. On Line 266, the authors mention that two of the MARV-exposed animals were positive for MARV RNA on Day 3; however, on Line 515, they state that only one animal was positive on Day 3. Please correct this discrepancy.

7. In Figures 1-4, data from the MARV-exposed animals are represented with the colour red, while data from the control animals are represented with the colour blue. In Figures 5, 7, S1, and S2, the opposite is true. Please revise the figures so that the colour coding is consistent.

8. Please indicate statistically significant differences between MARV-exposed and control animals with an asterisk (or asterisks) in Figures 3, 4, 7, S1, and S2. This would add important context to the figures that otherwise has to be fished out of the text.

9. For all other parameters analyzed, the authors report the data from the terminal time points separately. The authors should consider doing the same for Figure 2.

10. The terminal data point in Figure 3 should be described in the text of the manuscript.

11. Please clarify whether arithmetic or geometric means are depicted in Figures S2a and 7.

12. “GE/ul” and “GEq/ul” are used interchangeably throughout the paper. Please revise the figures and text for consistency.

13. Lines 67-69: Consider revising this sentence. Currently, the sentence implies that MARV disease is 100% lethal in both NHPs and humans.

14. Line 208: “underlying mucle” should be “underlying muscle”.

15. Line 209: “jejenum” should be “jejunum”.

16. Line 506: To be consistent with other parts of the text, replace “120 hours” with “5 days”.

Round 2

Reviewer 2 Report

Comments were all addressed to my satisfaction. Very nice paper!